# Identification of Novel Peptides with Alcohol Dehydrogenase (ADH) Activating Ability in Chickpea Protein Hydrolysates

**DOI:** 10.3390/foods12081574

**Published:** 2023-04-07

**Authors:** Rong Zan, Ling Zhu, Gangcheng Wu, Hui Zhang

**Affiliations:** 1School of Food Science and Technology, Jiangnan University, Wuxi 214122, China; 2National Engineering Research Center for Functional Food, Collaborative Innovation Center of Food Safety and Quality Control in Jiangsu Province, Jiangnan University, Wuxi 214122, China

**Keywords:** chickpea protein hydrolysates, bioactive peptides, gastrointestinal digestion, molecular docking

## Abstract

Alcohol dehydrogenase (ADH) is one of the main rate-limiting enzymes in alcohol metabolism. Food protein-derived peptides are thought to have ADH activating ability. We verified for the first time that chickpea protein hydrolysates (CPHs) had the ability to activate ADH and identified novel peptides from them. CPHs obtained by hydrolysis with Alcalase for 30 min (CPHs-Pro-30) showed the highest ADH activating ability, and the ADH activation rate could still maintain more than 80% after in vitro simulated gastrointestinal digestion. We have verified four peptides with activation ability to ADH: ILPHF, MFPHLPSF, LMLPHF and FDLPALRF (concentration for 50% of maximal effect (EC_50_): 1.56 ± 0.07 µM, 1.62 ± 0.23 µM, 1.76 ± 0.03 µM and 9.11 ± 0.11 µM, respectively). Molecular docking showed that the mechanism for activating ADH was due to the formation of a stable complex between the peptide and the active center of ADH through hydrogen bonding. The findings suggest that CPHs and peptides with ADH activating ability may be developed as natural anti-alcoholic ingredients to prevent alcoholic liver disease (ALD).

## 1. Introduction

The human liver contains a variety of alcohol metabolism pathways, among which the catalytic system triggered by alcohol dehydrogenase (ADH) is one of the most critical pathways [1], where ADH and acetaldehyde dehydrogenase (ALDH) act as the main rate-limiting enzymes [2]. ADH is a zinc-containing metalloenzyme with broad specificity. When activated and reacting with nicotinamide adenine dinucleotide (NAD+), ADH and ALDH convert about 90% of the consumed alcohol in this pathway into acetaldehyde and acetic acid [3], with most of the acetic acid transferred to blood and extrahepatic tissues [4]. However, the metabolic process of alcohol in the liver may lead to alcohol toxicity [5]. Short-term excessive drinking can reduce the activity of ADH in the liver, and excessive accumulation of alcohol will cause alcoholic liver disease (ALD) [6]. Aside from abstaining from alcohol, there are currently no effective medications for ALD [7]. As a result, there is growing interest in finding safe and effective natural ingredients, especially food protein, that can accelerate alcohol metabolism and prevent alcoholic liver disease.

Chickpea (*Cicer arietinum* L.) is a nutritionally superior plant protein source of complete protein [8]. However, research on its ADH activating ability and the identification of ADH activating peptide sequences from it is lacking. Most previous studies have focused on the ability of protein hydrolysates to activate ADH, but have not investigated whether the ADH activating ability of protein hydrolysates can be maintained after gastrointestinal digestion. Currently only two studies have identified, synthesized and validated peptides with activation ability to ADH. One study identified two peptides from mushroom foot protein [1], while the other identified six peptides from crucian carp swimming bladder [9].

This study investigated a new biological activity of chickpea protein source hydrolysates, namely their ability to activate ADH. For the first time, we explored the ADH activation ability of CPHs, optimized the enzymatic hydrolysis time, and evaluated the stability of CPHs after in vitro gastrointestinal simulated digestion. Finally, we identified the ADH activating peptide sequences of CPHs.

## 2. Materials and Methods

### 2.1. Materials

The chickpea used in this study was the kabuli variety produced in Mulei County (Xinjiang, China) and was obtained from Amway (Shanghai) Technology Development Co., Ltd. (Shanghai, China). Alcalase, Protamex and Neutrase were purchased from Novozymes (Beijing, China). Papain was provided by Pang Bo Bioengineering Co., Ltd. (Nanning, China). Pepsin and Trypsin were purchased from Sigma-Aldrich (Shanghai, China). The ADH detection kit was purchased from Nanjing Jiancheng Bioengineering Institute (A083-1-1, Nanjing, China). All the other reagents used in the experiment were of analytical grade.

### 2.2. Extraction of Chickpea Protein

To extract chickpea protein, the chickpeas were crushed, and the undersize was collected through a 40-mesh sieve. The alkali-dissolving and acid-precipitating method was used to extract the protein. Chickpea flour and water were mixed according to the ratio of 1:12 (*w*/*v*), and the mixture was stirred for 1.5 h at 25 °C to extract protein while keeping the pH at 11.0. After centrifugation at 2140 g for 20 min, the supernatants were collected. The precipitate and water were mixed according to the ratio of 1:5 (*w*/*v*). The same extraction steps were repeated twice. The pH of the supernatants was adjusted to 4.0 before centrifugation, and the precipitate was washed twice with pH 4.0 HCl water to remove starch. Then, the precipitate was redissolved in distilled water after centrifugation again. The chickpea protein powder was obtained after desalination by dialysis and freeze drying. To remove lipids and prevent the formation of an emulsified phase [10], the chickpea protein powder was degreased twice using n-hexane in a 1:5 (*w*/*v*) ratio and stored at 4 °C for future use. Delipidation not only improves the recovery of the peptide [10] but also changes the structure of lipoproteins, which could be a valuable source of bioactive peptides.

### 2.3. Preparation of CPHs

Chickpea protein aqueous solution (4%) was subjected to enzymatic hydrolysis using four different proteases. The hydrolysis was carried out at the optimum temperature and pH of each protease after being placed in a water bath at 80 °C for 20 min with an enzyme-to-substrate ratio of 8000 U/g. The optimal conditions for enzymatic hydrolysis using the four proteases are as follows: Alcalase: 55 °C, pH 8.5; Neutrase: 45 °C, pH 7.0; Protamex: 55 °C, pH 8.0; Papain: 55 °C, pH 7.0. The pH was immediately adjusted to 7.0 after the enzymatic hydrolysis, and the enzymatic solution was placed in a boiling water bath for 10 min to inactivate the protease. CPHs were obtained by freeze-drying the supernatant and stored at 4 °C for future use.

### 2.4. Determination of the Degree of Hydrolysis (DH)

The DH was determined by OPA method [11]. The mixture consisted of 400 μL sample solution (1 mg/mL) and 3 mL OPA, which was mixed well and stayed for 2 min. The absorbance value (A_340_) was recorded at 340 nm. Distilled water or serine standard solution (0.1 mg/mL) was used as blank solution and standard solution. The formula for determining DH (%) is as follows:(1)$$\mathrm{serine}{\mathrm{NH}}_{2}=\frac{A_{\mathrm{sample}}-A_{\mathrm{blank}}}{A_{\mathrm{standard}}-A_{\mathrm{blank}}}\times 0.9516\times \frac{0.1}{m\times P_{\mathrm{sample}}}\times 100$$
(2)$$\mathrm{DH}(\%)=\frac{\mathrm{serine}\;\mathrm{NH}2-\beta }{\alpha }\times \frac{1}{\mathrm{ht}ot}\times 100$$
where serine NH_2_ (mM/g) is the average amino content; A_standard_, A_sample_ and A_blank_ denote the absorbance values of the standard, sample and blank control, respectively; 0.9516 (mM) is the concentration of serine standard solution; 0.1 (L) is the volume of the sample; m (g) is the mass of the sample; P_sample_ (g/100 g) is the protein content of the sample; α and β are correction factors, regarded as constants (α and β of chickpea protein are 1.00 and 0.40, respectively). h_tot_ (meqv/g) is the number of peptide bonds (h_tot_ of chickpea protein is 7.22 meqv/g).

### 2.5. Determination of ADH Activation Rates In Vitro

We referred to the experimental method of Shi et al. [9], and the in vitro activity was determined according to the method of ADH detection kit. The sample solution (50 μL) was mixed with the working solution (150 μL). After the incubation at 37 °C for 5 min, ADH solution (0.2 U/mL, 50 μL) was added for reaction. The absorbance value (A_340_) was measured at 340 nm every minute for 10 min. In the background group, distilled water was used instead of ADH solution. The control group replaced the sample solution with distilled water. In the blank group, distilled water was used instead of ADH solution and sample solution. The formulas for calculating initial reaction rate and ADH activation rate are as follows:Initial reaction rate = ∆A_340_/∆min(3)
(4)$$\mathrm{ADH}\;\mathrm{activation}\;\mathrm{rate}\;(\%)=\frac{\left(V_{\mathrm{sample}}-V_{\mathrm{background}})-(V_{\mathrm{control}}-V_{\mathrm{blank}}\right)}{V_{\mathrm{control}}-V_{\mathrm{blank}}}\times 100\%$$
where V_sample_, V_background_, V_control_ and V_blank_ denote the initial reaction rates of the sample, background, control and blank groups, respectively.

### 2.6. In Vitro Simulated Gastrointestinal Digestion

Determination of in vitro simulated gastrointestinal digestion followed the method of Brodkorb et al. [12]. A solution of enzymatic hydrolysis was prepared by dissolving 10 g CPHs in 100 mL of 2000 U/mL Pepsin hydrochloric acid solution (pH 3.0) and was cultured in a constant temperature shaking incubator at 37 °C for 2 h (150 rpm) to simulate gastric digestion (SGD). Afterwards, the pH of the digestive system was immediately adjusted to 7.0 with 1 mol/L NaHCO_3_, and the volume was fixed to 200 mL for simulated intestinal digestion (SID). The solution (100 mL) was inactivated in a water bath at 80 °C for 10 min and rapidly cooled, and this time was recorded as 2 h. Trypsin (100 U/mL) was added to the remaining samples, and samples were taken at different time points under the above digestion time to inactivate the protease and cooled rapidly. The samples were then subjected to subsequent ADH activation assay. The control group consisted of CPHs without simulated digestion in vitro.

### 2.7. Peptide Identification and Synthesis

The peptide sequences in CPHs were identified by Shanghai Omicsolution Co., Ltd. (Shanghai, China). The desalted CPHs were dissolved in solvent A (0.1% formic acid aqueous solution). The reaction process was analyzed by Q Exactive^TM^ coupled to the EASY-nanoLC 1200 system (Thermo Fisher Scientific, Waltham, MA, USA). Tandem mass spectra were analyzed by PEAKS Studio version 10.6 (Bioinformatics Solutions Inc., Waterloo, ON Canada). PEAKS DB was built to search the database of uniprot_Cicer arietinum (version 202112, 24812 entries) assuming none as the digestion enzyme. The filtered proteins had −10 lgP ≥ 0, and the peptides had −10 lgP ≥ 20 and contained at least one unique peptide.

### 2.8. Computer-Aided Screening of Peptide Sequences

Peptide sequences were screened from the assay results. The biological activity prediction score of peptides was queried through PeptideRanker (http://distilldeep.ucd.ie/PeptideRanker/ (accessed on 14 August 2022)). The protein source of peptides was queried through the protein database (https://www.uniprot.org/ (accessed on 29 August 2022)). The toxicity prediction of peptides was queried through Toxinpred (https://crdd.osdd.net/raghava/toxinpred/ (accessed on 1 September 2022)). The sensitization prediction of peptides was queried through AllerTOP (https://www.ddg-pharmfac.net/AllerTOP/ (accessed on 1 September 2022)).

### 2.9. Molecular Docking

The crystal structure of yeast ADH (PDB ID: 5ENV) was obtained from the PDB database (http://www.rcsb.org/pdb (accessed on 15 September 2022)). The 2D and 3D structures of the peptides were constructed using Chem Draw (ChemBio Draw 14.0.0.11, Waltham, Massachusetts, USA) and Chem 3D (ChemBio 3D 14.0.0.117, Waltham, Massachusetts, USA). The original ADH ligand and water molecules were removed using PyMOL (PyMOL Molecular Graphics System 2.2.0, New York City, New York, USA). ADH and peptides were pretreated using AutoDockTools (AutoDockTools 1.5.6, La Jolla, California, USA). The docking energy of the enzyme active site was obtained by analyzing the results of molecular simulation docking. The lowest docking energy corresponded to the most stable molecular docking structure. The visualization of molecular docking results was handled with PyMOL.

### 2.10. Solid-Phase Synthesis of the Peptides

Peptides (purity > 95%) were synthesized and provided by ChinaPeptides Co., Ltd. (Wuhan, China) using Fmoc solid-phase synthesis.

### 2.11. Statistical Analysis

One-way analysis of variance and SPSS statistics were used for data processing. All assays were performed and repeated in three independent experiments. Results were expressed as mean ± standard deviation. Differences in results with *p* < 0.05 were considered statistically significant.

## 3. Results and Discussion

### 3.1. Screening of Proteases

The hydrolysis effects of four proteases and the activation effects of CPHs on ADH are shown in Figure 1A. It can be observed that the sequences of CPHs under different protease hydrolysis conditions were different. Among the four proteases, Alcalase hydrolysis produced CPHs with the highest activation activity on ADH (68.30 ± 2.78%, the actual reaction concentration was 0.2 mg/mL), followed by Papain, Protamex and Neutrase. The ADH activation rate of CPHs obtained by Alcalase was significantly higher than that of CPHs obtained by other three proteases.

It is evident that Alcalase is the most effective protease to convert chickpea protein into potential peptides, as it could release more peptides to activate ADH. CPHs obtained by Papain hydrolysis showed higher DH, which might be due to the internal cleaving of the polypeptide chain, away from the end [13]. The concentration for 50% of maximal effect (EC_50_) value of CPHs produced by Alcalase hydrolysis was 0.0723 ± 0.0022 mg/mL (Figure 1B), which was significantly lower than that of 6.237 ± 0.795 mg/mL for the low molecular weight peptides (0–3 kDa) of mushroom foot peptides [1]. Therefore, Alcalase may be an optimized protease to prepare CPHs with strong ADH activating ability.

### 3.2. Optimization of Time

The typical hydrolysis curve and DH change curve reflected the effect of the hydrolysis time on the hydrolysis process of chickpea protein. The effect of hydrolysis time on the hydrolysis process and the activation ability of hydrolysates on ADH are shown in Figure 2A. The ADH activation of CPHs (the actual reaction concentration was 0.2 mg/mL) showed an overall trend of increasing first and then decreasing within 4 h of hydrolysis, and the activation rate reached the highest at 120 min (69.48 ± 0.82%). Although the ADH activation rate at 120 min was higher than that at 30 min (66.45 ± 1.46%), there was no significant difference in their values, and the inhibition rate at this point was significantly different from other time points (*p* < 0.05). Based on these results, CPHs obtained by Alcalase hydrolysis exhibited the highest ADH activation at an optimal enzymatic hydrolysis time of 30 min (CPHs-Pro-30). The EC_50_ value of CPHs-Pro-30 on ADH activation was 0.0569 ± 0.0083 mg/mL (Figure 2B), which was significantly lower than that of 5.967 ± 0.776 mg/mL for the mushroom foot peptide P-1 fraction [1]. These findings indicate that CPHs-Pro-30 has an extremely significant activation on ADH and is worthy of further study.

### 3.3. Gastrointestinal Digestion Stability of CPHs-Pro-30

The activity will be reduced to a certain extent during the process of gastrointestinal digestion and absorption by the small intestine [14]. In vitro simulated gastrointestinal digestion is often used to evaluate the resistance of protein hydrolysates or peptides to gastrointestinal degradation [15,16]. The retention of the ADH activation ability of CPHs-Pro-30 after in vitro digestive stimulation is shown in Figure 3. The results showed that after CPHs-Pro-30 underwent SGD for 120 min, the ADH activation rate decreased from 68.25 ± 1.19% to 67.10 ± 1.69%, and the activity remained at 98.32%. As reported by Jang et al. and Escudero et al., Pepsin digestion did not significantly affect the biological activity of shark peptides and ham peptides, which might be due to the peptides having a certain resistance to gastric digestion or peptides being degraded into peptides with a smaller molecular weight that were also bioactive [17,18]. The ADH activation rate of CPHs-Pro-30 after SID decreased from 67.10 ± 1.69% to 56.41 ± 0.24%, and the activity remained at 84.07%. Trypsin digestion reduces the overall hydrophobicity of protein hydrolysates, resulting in altered bioactivity [19]. In addition, high hydrophobicity is a key factor for maintaining high ADH activity [20]. Therefore, the digestion by Trypsin leads to a reduction in the overall hydrophobicity of the polypeptide, and the degradation of certain hydrophobic peptides can result in the loss of activity. The ADH activation rate showed a decreasing trend after in vitro simulated digestion. Notably, the decrease in activation ability primarily occurred during the intestinal digestion stage, and the hydrolysis of peptides by Trypsin was found to be unfavorable for retaining peptides with high activation ability. Despite this, the ADH activation rate of CPHs-Pro-30 remained at 82.65% at the end of the in vitro digestion process, indicating that CPHs-Pro-30 has significant resistance to gastrointestinal digestion in vitro and warrants further investigation.

### 3.4. Identification by Peptidomics and Screening of CPHs-Pro-30

CPHs were identified by peptidomics. Sequences not modified by functional groups were selected, had a peptide chain length < 15 and peak area > 1.00 × 10^7^, belonged to protein sources rather than other enzyme sources, had an activity score > 0.8, were non-toxic, non-allergenic and amphipathic, had steric hindrance ≤0.65, hydrophobic amino acid ratio > 55% and characteristics amino acid ratio > 55% (characteristic amino acids: amino acids commonly contained in peptides with ADH activity reported in the literature). Wherein, hydrophobic amino acids include: L, A, I, P, F, V, W, Y, M [21], and characteristic amino acids include: L, A, I, P, F, Y, V, E [1,5,22,23,24]. Peptides with such amino acids may activate ADH.

After applying all the screening conditions mentioned above, a total of 18 novel potential bioactive peptides were identified (FDLPALR, FDLPALRF, FDLPALRW, FLRF, IDFEPFRP, IFVPHW, IFYVPRYFP, ILPHF, ILPHFF, IRFL, KFL, LFR, LLPHF, LLRF, LMLPHF, LRFL, MFPHLPSF and SFDLPALRF). The physicochemical properties and activity scores of these 18 peptides are summarized in Table 1. Furthermore, molecular docking was employed to investigate the autonomous binding ability of these peptides to ADH in order to evaluate the efficacy of the previous screening process.

### 3.5. Validation through Molecular Docking

In order to study the mechanism of the above 18 peptides on ADH activation at the molecular level, we carried out molecular docking of each peptide with ADH (Figure 4). By studying the docking of peptides and ADH, the study of the formed complex is beneficial to the screening of active peptides [25]. According to the binding energy as the evaluation standard, we found that the binding energies of these peptides with ADH were all negative (−7.60–(−10.26) kcal/mol, Table 1). This indicated that all of them could autonomously bind to ADH and may have ADH activating ability. A lower binding energy means the peptide is more stable in the active pocket of ADH [26]. The binding energy (−9.47 kcal/mol) of GLpGER reported by Shi et al. [9] was close to our results. Among them, ILPHF, MFPHLPSF, LLRF, FDLPALRF and LMLPHF had lower binding energy (<−9.60 kcal/moL) than other peptides. This indicated that their complexes with ADH were more stable than other peptides and might have stronger ADH activating ability, which would be verified in subsequent ADH activation assays.

ADH is a Zn-containing metalloenzyme with two subunits, one is located in the active center, and the other plays a role in stabilizing the quaternary structure. We selected the five peptides with the lowest binding energy to visualize the results of molecular docking. The number of hydrogen bonds and hydrogen bonding sites of these five peptides combined with ADH are shown in Table 2 (sorted by binding energy from low to high); the interaction models are shown in Figure 4A–E. Twelve hydrogen bonds were formed between ILPHF and ADH residues (THR45, HIS44, ARG340, ILE337, LEU182, VAL245, GLY177, ASP201, MET332 and LYS206). Seven hydrogen bonds were formed between MFPHLPSF and ADH residues (HIS48, LEU182, GLY181, ASP201, LYS206 and VAL247). Seven hydrogen bonds were formed between LLRF and ADH residues (VAL245, GLY177, GLY183, LEU182, GLY335, MET332 and ILE337). Eight hydrogen bonds were formed between FDLPALRF and ADH residues (THR45, VAL247, GLY339 and GLU333). Nine hydrogen bonds were formed between LMLPHF and ADH residues (HIS44, GLY181, VAL245, GLY335, LYS334 and GLU333). Notably, FDLPALRF and ILPHF formed hydrogen bonds with ADH amino acid residue THR45. ILPHF and LMLPHF formed hydrogen bonds with amino acid residue HIS44. MFPHLPSF formed a hydrogen bond with amino acid residue HIS48. Lastly, THR45, HIS44 and HIS48 could be connected to the Zn atom (one of the active sites of ADH) through the amino acid chain of HIS48→LEU47→ASP46→THR45→HIS44→CYS43→Zn atom.

Based on the results of peptidomics screening and molecular docking verification, ILPHF, MFPHLPSF, LLRF, FDLPALRF and LMLPHF were identified as potential ADH activators. These peptides were found to bind directly or indirectly to the active center or hydrophobic cavity of ADH. To further verify their potential as ADH activators, these five peptides were synthesized and tested in vitro.

### 3.6. In Vitro Activity Verification of Peptides

The above five peptides with potential activating ability to ADH were synthesized, and their in vitro activating activity was determined (Table 3). Among them, except for LLRF, all the other four peptides had ADH activation ability. In contrast, ILPHF, MFPHLPSF and LMLPHF had low EC_50_ values, exhibited very prominent activity and were the best ADH activators. The EC_50_ values were 1.56 ± 0.07 mM, 1.62 ± 0.23 mM and 1.76 ± 0.03 mM, which were significantly lower than that reported by Zhao et al. [1] for IPLH and IPIVLL (EC_50_: 7.15 mM, 8.62 mM, respectively). The EC_50_ values indicated that ILPHF, MFPHLPSF and LMLPHF had extremely significant activation effects on ADH. Meanwhile, FDLPALRF showed relatively low ADH activating ability with an EC_50_ value of 9.11 ± 0.11 mM. It is worth noting that there was no significant differences among the EC_50_ values of the three peptides, and they were consistent with their binding energy ranking for molecular docking, which verified the molecular docking results. That is, the lower the molecular docking energy, the smaller the EC_50_ value of the peptide. Hence, ILPHF, MFPHLPSF, LMLPHF and FDLPARF formed stable complexes with the active center of ADH through combined hydrogen bonds, thereby activating the enzymatic function of ADH.

## 4. Conclusions

The study found that CPHs-Pro-30 had the highest ADH activating ability (EC_50_: 0.0569 ± 0.0083 mg/mL) and exhibited remarkable resistance to in vitro gastrointestinal digestion. Through the identification of peptidomics, screening of peptide sequences and validation of molecular docking, five novel peptides (ILPHF, MFPHLPSF, LMLPHF, FDLPALRF and LLRF) with potential ADH activation ability were identified. All of them could be closely combined with the active center of ADH through combined hydrogen bonds. The results of in vitro activity verification showed that all four peptides (ILPHF, MFPHLPSF, LMLPHF and FDLPALRF) exhibited varying degrees of the ADH activating ability. Among them, the ADH activating ability of ILPHF was the most prominent (IC_50_: 1.56 ± 0.07 μM), followed by MFPHLPSF, LMLPHF and FDLPALRF (IC_50_: 1.62 ± 0.23 μM, 1.76 ± 0.03 μM, 9.11 ± 0.11 μM, respectively). Hence, we demonstrated that chickpea protein is an excellent source of peptides with high activation ability to ADH, and the peptides released from chickpea protein are promising to intervene in ALD by activating ADH.

## Figures and Tables

**Figure 1 foods-12-01574-f001:**
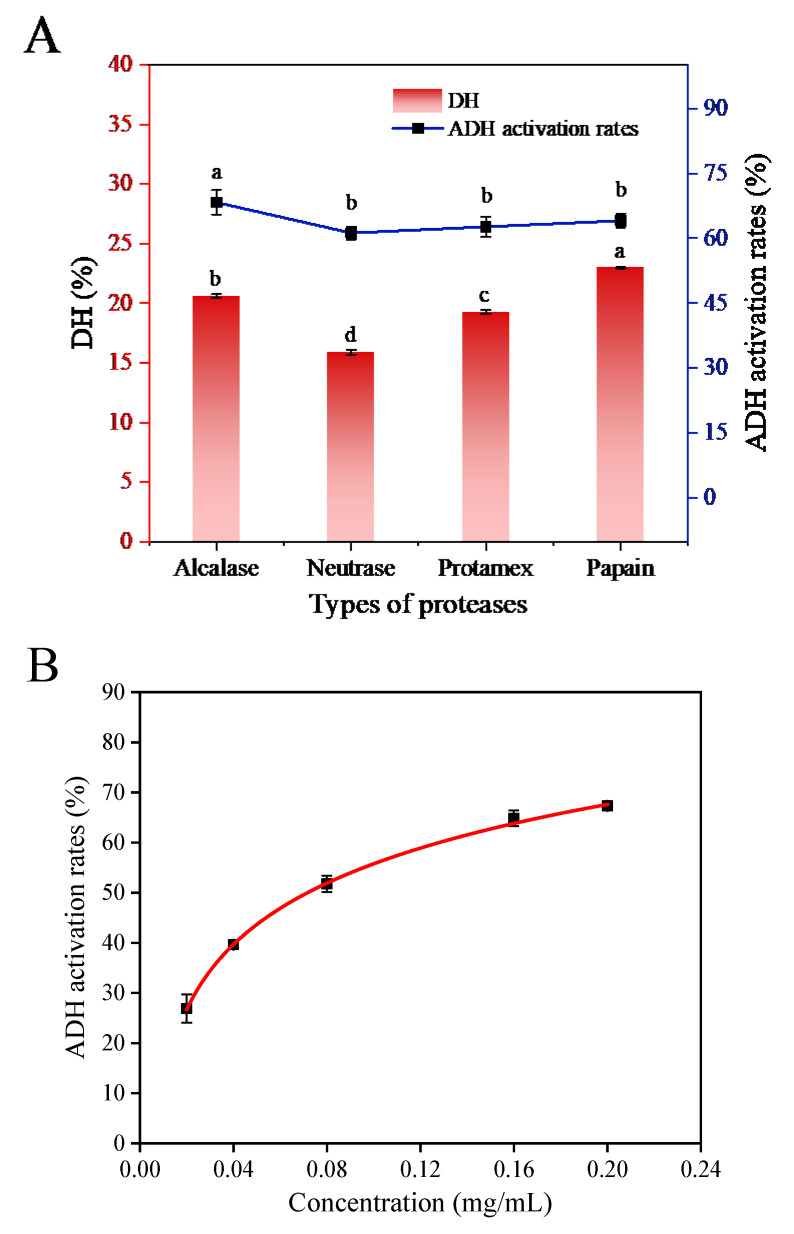
(**A**) ADH activation rates and DH of different proteases on hydrolysis of chickpea. (**B**) Concentration for 50% of maximal effect (EC50) values for ADH activation by CPHs. (Different letters (a–d) represent significant differences in values).

**Figure 2 foods-12-01574-f002:**
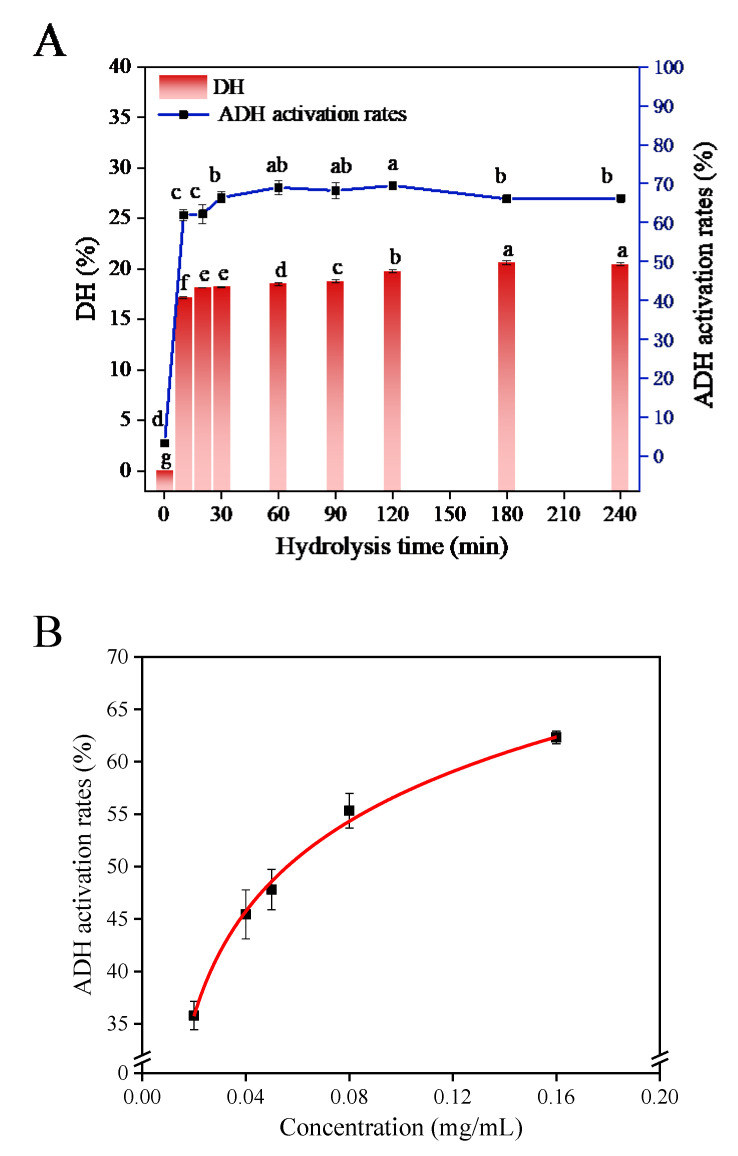
(**A**) ADH activation rates and DH of Alcalase in hydrolysis time. (**B**) EC_50_ values for ADH activation by CPHs-Pro-30. (Different letters (a–g) represent significant differences in values.)

**Figure 3 foods-12-01574-f003:**
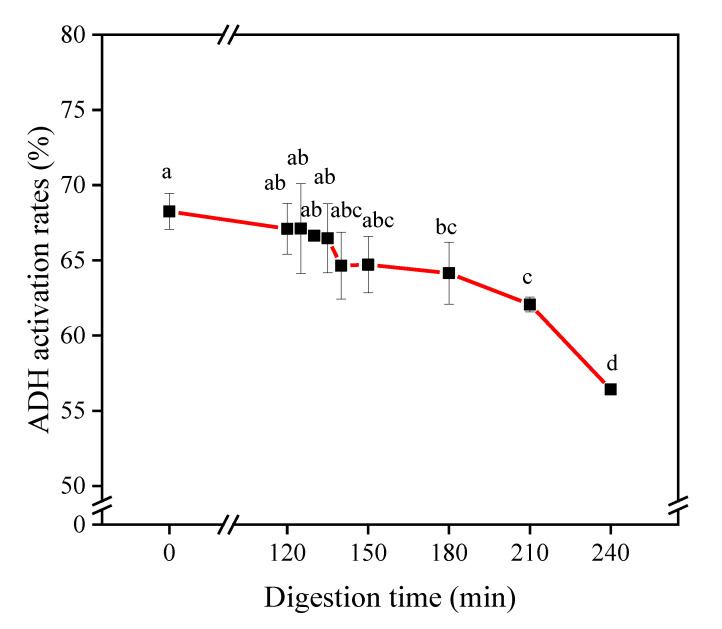
ADH activation rates of CPHs-Pro-30 at different concentrations after simulated gastric digestion (SGD) or simulated intestinal digestion (SID). (Different letters (a–d) represent significant differences in values).

**Figure 4 foods-12-01574-f004:**
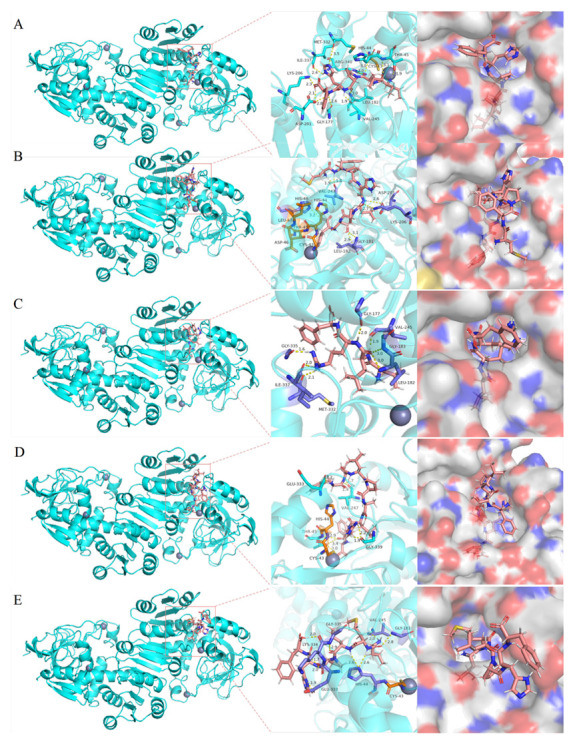
Molecular docking diagram of synthetic peptides (**A**) ILPHF, (**B**) MFPHLPSF, (**C**) LLRF, (**D**) FDLPALRF and (**E**) LMLPHF with ADH. (The 3D pink stick models are peptides, the purple stick models are ADH residues, the orange stick models are ADH residues directly linked to the Zn atom and the gray spherical model is Zn atoms. Hydrogen bonds are shown with yellow dotted lines.)

**Table 1 foods-12-01574-t001:** Physiochemical characteristics, activity scores and binding energy with ADH from synthetic peptides.

Serial Number	Sequence	Length	Relative Molecular Mass (Da)	Activity Score	Steric Hindrance	Amphipathic	Hydrophobic Amino Acid Ratio (%)	Characteristic Amino Acid Ratio (%)	Binding Energy (kcal/mol)
1	FDLPALR	7	831.07	0.82	0.58	0.35	71.43	71.43	−9.26
2	FDLPALRF	8	978.26	0.94	0.60	0.31	75.00	75.00	−9.87
3	FDLPALRW	8	1017.30	0.93	0.57	0.31	75.00	62.50	−8.69
4	FLRF	4	581.76	0.99	0.65	0.61	75.00	75.00	−9.58
5	IDFEPFRP	8	1020.25	0.86	0.62	0.47	62.50	75.00	−9.56
6	IFVPHW	6	798.04	0.86	0.49	0.24	83.33	66.67	−9.44
7	IFYVPRYFP	9	1201.55	0.83	0.62	0.27	88.89	77.78	−8.55
8	ILPHF	5	625.84	0.83	0.46	0.29	80.00	80.00	−10.26
9	ILPHFF	6	773.03	0.94	0.50	0.24	83.33	83.33	−8.46
10	IRFL	4	547.75	0.88	0.65	0.61	75.00	75.00	−8.08
11	KFL	3	406.56	0.83	0.64	1.22	66.67	66.67	−9.23
12	LFR	3	434.57	0.94	0.64	0.82	66.67	66.67	−9.14
13	LLPHF	5	625.84	0.88	0.42	0.29	80.00	80.00	−9.10
14	LLRF	4	547.75	0.90	0.61	0.61	75.00	75.00	−9.97
15	LMLPHF	6	757.05	0.86	0.48	0.24	83.33	66.67	−9.66
16	LRFL	4	547.75	0.91	0.61	0.61	75.00	75.00	−9.57
17	MFPHLPSF	8	975.28	0.93	0.49	0.18	75.00	62.50	−10.21
18	SFDLPALRF	9	1065.35	0.92	0.59	0.27	66.67	66.67	−7.60

**Table 2 foods-12-01574-t002:** Number of hydrogen bonds and binding sites from peptides.

Serial Number	Sequence	Number	Binding Sites
1	ILPHF	12	THR45, HIS44, ARG340, ILE337, LEU182, VAL245, GLY177, ASP201, MET332, LYS206
2	MFPHLPSF	7	HIS48, LEU182, GLY181, ASP201, LYS206, VAL247
3	LLRF	7	VAL245, GLY177, GLY183, LEU182, GLY335, MET332, ILE337
4	FDLPALRF	8	THR45, VAL247, GLY339, GLU333
5	LMLPHF	9	HIS44, GLY181, VAL245, GLY335, LYS334, GLU333

**Table 3 foods-12-01574-t003:** ADH activation concentration for 50% of maximal effect (EC_50_) of synthetic peptides in vitro. Different letters (a, b) represent significant differences at *p* < 0.05.

Serial Number	Sequence	ADH Activation EC_50_ ^a^ (mM)
1	ILPHF	1.56 ± 0.07 ^b^
2	MFPHLPSF	1.62 ± 0.23 ^b^
3	LMLPHF	1.76 ± 0.03 ^b^
4	FDLPALRF	9.11 ± 0.11 ^a^

Different letters (^a^,^b^) represent significant differences in values.

## Data Availability

Data are contained within the article.

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
