# Peer review of "Identification of Novel Peptides with Alcohol Dehydrogenase (ADH) Activating Ability in Chickpea Protein Hydrolysates"

_foods, 2023, doi:10.3390/foods12081574_

Round 1
Reviewer 1 Report
Review report for Foods-2257760
Although topic is interesting and experimental design is valid, several problems exist with this manuscript. Main issue is quality of English that really needs to be address prior to publication. Secondly, graphic abstract is confusing, giving idea that digestion and molecular docking are done simultaneously. Importantly, some crucial details are missing from Materials and Methods section. Specific comments are listed below:
Abstract: Needs to be re-written. For example, two last sentences are actually conveying the same information. Also, the “activity to activate” should be substituted with “tendency (or ability) to activate; anti-alcoholic should be used instead of anti-drunk, etc.
Introduction:
Line 33: NAD needs to be defined.
Lines 35-36: “Alcohol absorption is mainly metabolized in the liver, which may cause alcohol toxicity”- this sentence makes no sense. Either adsorbed alcohol is metabolized or non-metabolized alcohol is adsorbed.
Line 42-43: What do you mean by “complete spectrum of amino acids”? All amino acids found in proteins?
Materials and Methods:
Line 66: What do you mean by “undersize”? Particles smaller than sieve pores?
Lines 71-72: What do you mean by “mixture was operated twice according to the above steps”? The extraction step was repeated 2 times?
Line 72: “should be adjusted” or were actually adjusted?
Line 77-78: Delipidation can also change structure of lipoproteins that could be valuable source of bioactive peptides.
Line 102-103: “We refered to the experimental method of Shi et al.”-this pertains to which step? Extraction or use of kit
Line 123: How was pH adjusted? Addition of bicarbonate or simply by NaOH? Because addition of NaOH could lead to further protein degradation and release of peptides that were NOT released during digestion.
Line 124: How was solution inactivated?
Line 125: Why was only trypsin added? This doesn’t faithfully represent digestion condition (no chymotrypsin, mucin, pancreatin or bile salts).
Reviewer 2 Report
I carefully read the manuscript Identification of Novel Peptides with Alcohol Dehydrogenase (ADH) activating ability in Chickpea Protein Hydrolysates by Zan et al. In general, it seems to me technically arduous work, with well-analyzed and presented results. Aspects to improve have to do with writing and interpretation. Here my notes:The introduction is very brief. A deeper introspection must be done on the subject. Although it is true that it is a pioneering study, more information should be noted on how the activation of this enzyme is governed. Furthermore, some information regarding the enzyme is welcome.
The results and discussion section is slightly speculative and not strictly based on the findings. For example, the differences, while significant in the degree of hydrolysis, enzyme activating activity (Fig 1) are numerically very similar and are referred to as a stark difference. The wording should change and present more fully what is shown in the figures. On the contrary, when referring to the results of the activation rate (EC50), the results are well contrasted with the literature.
In general, a little more discussion, of the mechanisms, of the biological explanations and not only instrumental ones is welcome.
I think that the section that refers to optimization is not entirely appropriate, since no experimental arrangement is used to find the optimum (which can be done in another investigation). Kinetics is what is presented and that is how it should be written.
Other minor details
L22
Hopefully should be changed to another word.
L44-45
What previous studies are you referring to?
L50-55
Please limit yourself to writing the purposes of the research and eliminate what may be a conclusion or even speculation.
L100-101
It is not clear what the chickpea protein factors represent.
L169
Why obviously? There is antecedent of this, or this expression is based on the later results. To rewrite
L212-214
It seems unnecessary or inappropriately worded. Rewrite or delete.
Round 2
Reviewer 1 Report
Please include all your responses to me in the main text.